# Countermovement, Hurdle, and Box Jumps: Data-Driven Exercise Selection

**DOI:** 10.3390/jfmk8020061

**Published:** 2023-05-11

**Authors:** M. Tino Janikov, Jan Pádecký, Valentin Doguet, James J. Tufano

**Affiliations:** 1Sport Sciences-Biomedical Department, Faculty of Physical Education and Sport, Charles University, 162 52 Prague, Czech Republic; tino.janikov@gmail.com (M.T.J.);; 2Atelier Maker^®^, 497 00 Doué-en-Anjou, France

**Keywords:** plyometric training, stretch-shortening cycle, exercise variation

## Abstract

Apart from squat jumps, countermovement jumps (CMJ), and drop jumps, differences among other jump variations are not as well researched, making data-driven exercise selection difficult. To address this gap, this study compared selected concentric and eccentric jump parameters of maximal effort CMJ, hurdle jumps over 50 cm hurdle (HJ), and box jumps onto a 50 cm box (BJ). Twenty recreationally trained men (25.2 ± 3.5 years) performed 3 repetitions of CMJs, HJs, and BJs, each on separate days. The data were collected using force platforms and a linear position transducer. The mean of 3 trials of each jump variation was analyzed using repeated measures ANOVA and Cohen’s d. Countermovement depth was significantly greater (*p* ≤ 0.05) and peak horizontal force significantly lower during CMJ compared to HJ and BJ. However, there were no differences in peak velocity, peak vertical and resultant force, and total impulsion time. Finally, BJ significantly decreased peak impact force by ~51% compared to CMJ and HJ. Therefore, the propulsive parameters of HJ and BJ seem to be similar to CMJ, despite CMJ having a greater countermovement depth. Furthermore, overall training load can be decreased dramatically by using BJ, which reduced peak impact force by approximately half.

## 1. Introduction

Plyometric training is widely used in strength and conditioning practice to increase power output [1]. The cornerstone of plyometric training involves movements that maximize the use of the stretch shortening cycle, which is defined as eccentric pre-stretch directly followed by an isometric amortization phase and explosive concentric muscle action [2] that results in augmented power expression compared to purely concentric explosive movements [3]. Although bodyweight countermovement jumps (CMJ) are commonly used lower body plyometric exercises [4,5], other variations, such as hurdle jumps (HJ) and box jumps (BJ), are also performed. Although these variations are often used interchangeably in practice, the nature of overcoming an obstacle like a hurdle or a box might result in changes to concentric and eccentric jump parameters, which may affect the resultant training adaptations over time. However, these jump types have not yet been compared in a single study; therefore, the magnitude of the differences are still unknown.

To the best of our knowledge, only two studies [6,7] have previously investigated these differences, but it is not yet possible to draw practical conclusions from their findings due to the following methodological shortcomings. The first study compared a single CMJ performed from a static standing position to the 3rd out of 4 continuous HJs over 4 hurdles [6]. The static nature of the CMJ and dynamic nature of the continuous HJs makes it difficult to attribute any differences between them to the specific jump type. Therefore, it is necessary to compare CMJ with HJ, where both variations are performed either as a single jump or as a sequence of continuous jumps. The second study reported no differences in propulsive parameters between jumps onto boxes of two different heights, as well as no relationship between maximal CMJ height and maximal achievable BJ height [7]. However, this study did not compare any other parameters between CMJ and BJ, leaving the differences between the jump variations largely unexplored.

Although some of the differences among CMJ, HJ, and BJ may seem intuitive, such as less impact force during BJ due to the lack of gravitational acceleration after reaching the apex of the jump [8], quantifying the magnitudes of these difference is important. As the exercise selection represents an important load management tool for coaches aiming to acutely decrease eccentric load (i.e., basketball, volleyball, etc.) while continuing structured concentric training, it would be desirable to base training prescriptions on data rather than solely on intuition.

Furthermore, some changes in jump technique may be present when jumping over or atop an obstacle compared to purely vertical jumping due to different direction and magnitudes of concentric forces as a result of horizontal movement [9,10]. Additionally, humans are likely to adjust their performance based on expected demands of the task at hand [11,12,13,14] (i.e., the expected impact upon landing or perceived fear of hitting the obstacle mid-flight), which might lead to differences in propulsive jump phases between the jump variations. Therefore, the purpose of the present study was to assess and quantify the differences among CMJ, HJ, and BJ in the same study in order to help guide coaching decisions. Based on the aforementioned points, we hypothesized that the BJ would result in less impact force than the other two variations. Furthermore, we hypothesize that subjects would jump with a greater countermovement depth in order to maximize the propulsion time during the HJ and BJ, thereby leading to greater forces and velocities in order to overcome the hurdle and the box when compared to the traditional CMJ.

## 2. Materials and Methods

### 2.1. Participants

Twenty recreationally trained university-aged men volunteered to participate in this study (25.2 ± 3.5 years, 180.2 ± 4.4 cm, 80.0 ± 7.8 kg, 11.5 ± 2.7% body fat, CMJ height 49.5 ± 6.4 cm). This sample size was shown to be appropriate by a-priori power analysis (effect size = 0.8; α err. prob. = 0.05; Power (1–β err. prob.) = 0.90) using G*Power 3.1.9.7 (RRID:SCR_013726), resulting in a necessary sample size of 19 subjects. The subjects had experience in sports where jumping was common, such as soccer, basketball, handball, track and field, and martial arts, but none of the subjects competed in any of these sports professionally. All subjects were able to perform countermovement vertical jumps and hold a light wooden dowel racked across the posterior shoulders without pain. The subjects reported no ongoing rehabilitation process post-injury or any other chronic conditions that could influence the results or prevent them from safely participating in this study. The subjects were asked to arrive rested (≥7 h of sleep and no exhausting lower body training 36 h before testing), well hydrated, and having fasted (≥2 h). Additionally, we asked the subjects to maintain their habitual dietary and supplementation intake while participating in the study. This study was completed in accordance with the Declaration of Helsinki. The experimental procedures were approved by the university Ethics Board (126/2018) and were explained to the subjects prior to providing institutionally approved written informed consent to participate in the study.

### 2.2. Data Collection

A quasi-randomized experimental approach was used to quantify the differences between the CMJ, HJ, and BJ in concentric and eccentric jump parameters. The experimental design is depicted in Figure 1. Every subject completed 3 laboratory visits separated by at least 48 h. All visits were scheduled at the same time of day (±1 h). The first visit began with measurements of body weight, body height, and body composition performed using an electronic column scale with a fitted stadiometer (Seca 769, Seca 220; Seca Ltd., Hamburg, Germany) and bioelectric impedance (InBody 720; Biospace Co., Ltd., Seoul, Republic of Korea), respectively. The subjects then completed a standardized warm up which included a single set per exercise of in-place running (30 ground contacts per leg), 10 in-place bilateral pogo jumps, dynamic unilateral stretches of hip, knee, and ankle muscles (3 repetitions per leg, per exercise), 10 bodyweight squats, followed by reverse lunges, side lunges, unilateral stiff-legged deadlifts, and supine lying unilateral hip raises for 5 repetitions without added load per leg for each exercise.

Next, to provide a specific warm-up and remove any possible effect of potentiation that could positively or negatively influence the experimental jumps, 10 preparatory CMJs without arm swing were performed, with the light wooden dowel held across their posterior shoulders behind the base of the neck. These jumps were performed using a running 10-s timer, allowing for roughly 8 s to prepare for the next jump. This jump frequency and volume were selected based on pilot testing, which showed no negative effects on following jumping performance, while allowing for a safe return to the starting position and preparation for the next jump in all 3 jump types. The total time necessary to complete the warm-up process was approximately 5 to 8 min.

After completing the warm-up, the subjects performed 3 maximal CMJs using the same jump technique and 10-s running timer as during the preparatory jumps. The subjects were instructed to jump as high as possible and to land softly while receiving verbal encouragement from the researchers. The countermovement depth, speed, and stance width during all jumps were self-selected by the subjects to maintain ecological validity. To conclude the first visit, the subjects were familiarized with the HJ and BJ procedures by performing a minimum of 3 repetitions of HJ and BJ. Additional repetitions were allowed if requested by a subject or considered necessary by the researchers.

At the beginning of the second and third visits, the subjects performed the same standardized warm-up, followed by 10 warm-up jumps, and 3 maximal jumps over the 50 cm hurdle or onto the 50 cm box with the wooden dowel held across the posterior shoulders as described above. The order of HJ and BJ was randomized. After the subjects jumped over the hurdle or onto the box, they stepped back to the starting position and prepared themselves to perform the next repetition at the end of the 10-s period. A visual depiction of the set-up including the athlete, linear position transducer, force plate(s), box, and hurdle is shown in Figure 2.

The ground reaction force data of all maximal jumps were recorded using two synchronized piezoelectric force plates (Kistler 9286BA; Kistler Instruments Inc., Winterthur, Switzerland). The force plates were placed side-by-side on the ground to measure CMJs. The first force plate, used to measure propulsive forces for both HJ and BJ, was positioned on the ground separated from the hurdle or box by 15 cm. The second force plate was used to measure landing forces. The position of the second force plate was on the ground 5 cm behind the hurdle or on top of the box 5 cm from the edge closest to the subject.

The ground reaction forces were recorded using sampling frequencies of 1000 Hz, a 16-bit A/D board amplifier, and BioWare v5.3.2.9 software (Kistler Instruments Inc., Winterthur, Switzerland). A custom MATLAB program (1.8.0.121; MathWorks, Natic, MA, USA) was used to calculate peak propulsive forces (PF) in vertical, horizontal, and resultant directions as maximal ground reaction force produced in each direction during the propulsive phase, total impulsion time as the time from the beginning of the countermovement to the take-off, and peak impact force as maximal resultant ground reaction force produced during the landing phase. A threshold of 20 N was used to identify individual phases of the jump.

The velocity and displacement data of all maximal jumps were recorded using a linear position transducer (GymAware Power Tool; Kinetic Performance Technology Pty. Ltd., Canberra, Australia). The string of the linear position transducer was attached to the wooden dowel 30 cm from the right shoulder to the end of the dowel [15]. The linear position transducer was placed on the ground directly below the end of the string for CMJ, and in the middle of the horizontal distance between the force plates in line with the attachment of the string for HJ and BJ. The attachment of the string and the position of the linear position transducer was selected due to the technical restrictions of the equipment used and the motion necessary to complete the measured task. However, the linear position transducer used in this study automatically corrects the data for horizontal displacement and allows the three conditions to be compared. The correct position of the string was checked before every jump. Additionally, one researcher observed the movement of the dowel during the data collection. Any trials with notable rotational dowel movements, deviation from horizontal dowel position, or other cases of failed trial (i.e., hitting the obstacle during the flight or not landing on the force plate) were excluded, and the trial was repeated after completing the prescribed rest interval. The internal software was used to calculate peak concentric velocity as maximal velocity of the dowel during the concentric propulsive phase, and countermovement depth as a maximum downward displacement of the dowel below the standing position during the propulsive phase.

### 2.3. Statistical Analysis

Data acquired during 3 repetitions of each jump type were averaged for each subject and used for analysis. The Shapiro-Wilk test and Quantile-Quantile plots were used to test the data for normality of distribution. Means and standard deviations (SDs) were calculated for all variables and repeated ANOVA measures were performed to assess the data. A Greenhouse-Geisser correction was used in instances where sphericity was not assumed. Pairwise comparisons were performed using a Holm-Bonferroni follow-up when appropriate. Alpha level for significance for all tests was set at ρ ≤ 0.05. Effect sizes (Cohen’s d) were calculated and interpreted as trivial (<0.20), small (0.20 to 0.49), moderate (0.50 to 0.79), and large (≥0.80) [16]. For the sake of clarity and concision, only moderate and large effect sizes will be discussed in further sections. All statistical analyses were performed using RStudio 2022.07.2 + 576 (Integrated Development Environment for R; RStudio, PBC; Boston, MA, USA). The countermovement depth, peak concentric velocity, and total impulsion time data are presented as absolute values; all ground reaction force data are presented as relative to the subject’s bodyweight (N·N^−1^). The intra-day intraclass correlation coefficients were calculated and interpreted as poor (<0.50), moderate (0.50 to 0.74), good (0.75 to 0.90), and excellent (>0.90) reliability [17].

## 3. Results

All variables have shown good to excellent reliability, with the only exception being moderate reliability of peak impact force during BJ (Figure 3, Figure 4, Figure 5 and Figure 6). Nevertheless, a total of five trials had to be discarded and repeated (3 CMJs, 1 HJ, 1 BJ) due to excessive dowel movement (4 trials) or missing the force plate upon landing (1 trial). No significant differences existed for PF-vertical, PF-resultant, peak concentric velocity, and total impulsion time (Figure 3 and Figure 4, respectively). However, a non-significant moderate effect was present for longer total impulsion time during CMJ compared to HJ (Figure 4b). There were significant moderate and small effects for deeper countermovement depth during CMJ compared to HJ and BJ, respectively (Figure 5a). Finally, significant large effects were shown for less PF-horizontal during CMJ compared to both HJ and BJ (Figure 5b), and for less peak impact force during BJ compared to both CMJ and HJ (Figure 6).

## 4. Discussion

The results of the present study show that the subjects adjusted the propulsive jump phase in response to the added obstacle during HJ and BJ compared to the CMJ condition. These adjustments were manifested as decreased countermovement depth and increased PF-horizontal. However, there were no significant differences in any of the other propulsive variables (PF-vertical, PF-resultant, peak concentric velocity, and total impulsion time). Furthermore, as expected, BJ significantly reduced peak impact force compared to HJ and CMJ conditions. These results partly support our initial hypothesis, but not fully, as the different jump types did not significantly differ in all measured variables.

This study’s most important finding is the lack of significant effects of jump type on PF-vertical, PF-resultant, and peak concentric velocity. Even though HJ and BJ conditions resulted in significantly larger PF-horizontal compared to CMJ, this approximately four-fold PF-horizontal difference was not sufficient to significantly influence the PF-vertical and PF-resultant. Similarly, a previous study reported non-significant trivial-to-small effects of box height for peak propulsive force, peak propulsive power, propulsive rate of force development, and concentric time to take-off [7]. This supports the possibility of interchangeably using CMJ, HJ, and BJ (from a propulsive perspective) in training practice, as similar magnitudes of propulsive forces and velocities likely impose similar training stimulus.

To maintain ecological validity, the subjects were allowed to adopt their preferred countermovement depth during each condition, which resulted in an interesting and unexpected finding. Previous studies have shown a relationship between countermovement depth and total impulsion time [18,19,20], and altering total impulsion time by manipulating countermovement depth can yield two different acute benefits. Increasing countermovement depth can allow for greater concentric work to be produced via greater available distance, which can also yield higher velocities [18,19,20,21,22]. Such exercise variations would find their place as sport-specific stimulus in athletes performing explosive motions from deep squat positions (i.e., ski jumps, Olympic weightlifting, swimming, sumo, etc.). On the other hand, decreasing the countermovement depth (i.e., decreasing the range of motion) and therefore decreasing the total impulsion time might allow for the more efficient utilization of the stretch-shortening cycle by increasing the rate of force development and power production in both eccentric and concentric jump phases [23], as well as increased eccentric work, amortization, and concentric force, and decreasing amortization time [24]. Therefore, exercise variations utilizing quicker jumps with smaller countermovement depth would be more specific in athletes taking advantage of the stretch-shortening cycle (i.e., basketball, volleyball, high jump, gymnastics, etc.).

Contrary to previous research, our results show significantly greater countermovement depth during CMJ compared to the other two included jump types, but this difference did not manifest in a significant difference in total impulsion time, PF-vertical, PF-resultant, or peak concentric velocity. The only moderate effect of jump type observed for these propulsive variables was a non-significantly shorter total impulsion time during HJ compared to CMJ. This discrepancy in comparison to the previous research might be a result of countermovement depth being measured as a downward shoulder displacement in our study. Therefore, smaller countermovement depth during HJ and BJ might be indicative of subjects keeping a slightly more upright torso position with the presence of the obstacle. However, the adjustments resulting from the added obstacle and leading to differences in countermovement depth and PF-horizontal were not large enough to meaningfully change the other dependent propulsive variables. Based on this, we consider HJ and BJ to be valid alternative variations to CMJ from the perspective of measured acute propulsive parameters.

As expected, there was a significant large effect of BJ for reducing peak impact force compared to HJ and CMJ. Our results are in line with previous research showing a significantly reduced sum of ankle, knee, and hip joint peak landing power during BJ compared to CMJ and HJ [8]. Nevertheless, the magnitude of peak impact force reduction during the BJ (~51%) was quite remarkable. Considering this, the main factors causing the reduction of peak impact force were probably a combination of instructions for a soft landing and a box height (50 cm) that coincidentally matched the mean jump height of the subjects (49.5 cm), causing decreased time for downward acceleration and subsequent lower landing velocity. Therefore, future research might take this a step further and evaluate the differences in landing forces during box-to-CMJ-height ratios other than the nearly 1:1 ratio that was present in this study.

Even though improving eccentric strength and landing mechanics are warranted in certain situations [25], other situations (i.e., periods of increased training and/or competitive load, or acute patellofemoral sensitivity) might require lowering impact force for load management [26,27]. For example, acute patellofemoral pain seems to decrease an athlete’s ability to effectively absorb impact force [28]. In turn, increased magnitudes and rates of eccentric forces developed in the patellofemoral joint during landing were significantly correlated with increased patellofemoral pain in young symptomatic women [29]. Therefore, according to our data, coaches can effectively use BJ to reduce peak impact force by ~51% compared to CMJ, while not decreasing concentric performance when decreased eccentric loading is warranted.

Although the present study provided data for evidence-based decision-making when prescribing three plyometric exercises, much is still unknown regarding this topic. For example, the effect of an enhanced stretch-shortening cycle by performing multiple continuous repetitions, fatigue resulting from a higher volume set, the effect of various box heights on IF, and kinematic analysis of the hip, knee, and ankle joint movements during the jump were not assessed in this study. Furthermore, this study had some limitations. Firstly, the attachment of the linear position transducer to one end of the dowel held by the subjects across the shoulders could potentially lead to some error of measurement if the subject’s shoulders tilted during the execution of the test. To mitigate the occurrence of this error, the movement of the dowel was observed by one researcher, and the trials where any notable rotational movements or deviation from horizontal position occurred were excluded. Secondly, jump technique (i.e., the amount of forward lean of the trunk) may to some extent influence the countermovement depth results, as the string of linear position transducer was attached to the dowel held across the shoulders. Therefore, future studies should aim to attach the string of the linear position transducer to a subject’s waist. Such attachment would allow them to use the arm swing while performing the jumps, which would be beneficial as the real-life training programs would likely not restrict the arm swing. However, this was not possible in the present study due to the specific technological constraints of equipment used during the HJ variation (i.e., string of the linear positional transducer colliding with the top of the hurdle upon landing). Therefore, we decided to use a string attached via the dowel in all jump types for consistency. Finally, studies investigating vertical jumps usually include jump height comparisons. However, the current study was not meant to assess height, but instead to compare important concentric and eccentric parameters influencing training adaptation. In this sense, jump height as an outcome would not have added much value to the discussion.

## 5. Conclusions

The present study provides data to support two distinct training considerations. Firstly, coaches can use a BJ with a box height similar to the maximum CMJ height to reduce an athlete’s peak impact force by approximately half, which can be highly beneficial during certain training periods where impact forces should be reduced. Secondly, coaches should be aware that CMJ results in a significantly greater countermovement of the shoulders compared to the BJ and HJ, which does not seem to provide any benefits for improving other propulsive parameters as no differences were seen in PF-vertical, PF-resultant, peak concentric velocity, or total impulsion time. Thus, a traditional CMJ may not be the ideal exercise choice for athletes who need to perform jumps quickly with minimal countermovement depth. On the other hand, there are other possible factors that might influence the exercise selection, so further research should be conducted in this area before coaches should definitely choose one jump type over another for propulsive training purposes.

## Figures and Tables

**Figure 1 jfmk-08-00061-f001:**
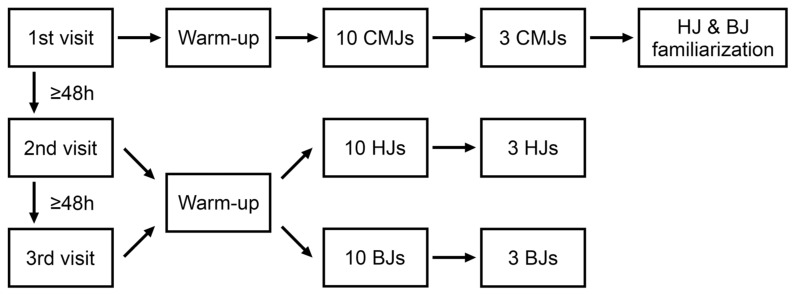
Experimental design.

**Figure 2 jfmk-08-00061-f002:**
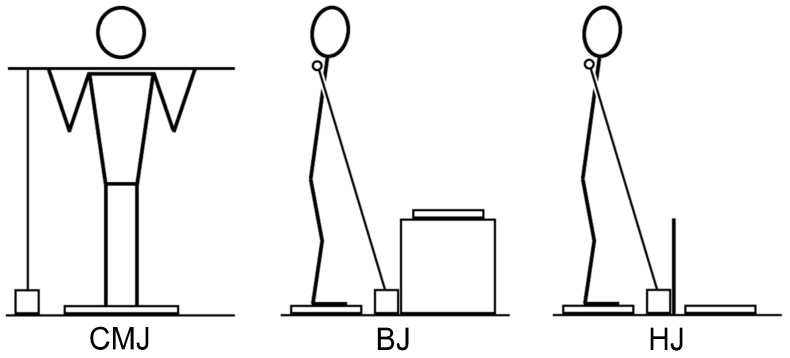
Data collection setup during the countermovement jump (CMJ), box jump (BJ), and hurdle jump (HJ) conditions.

**Figure 3 jfmk-08-00061-f003:**
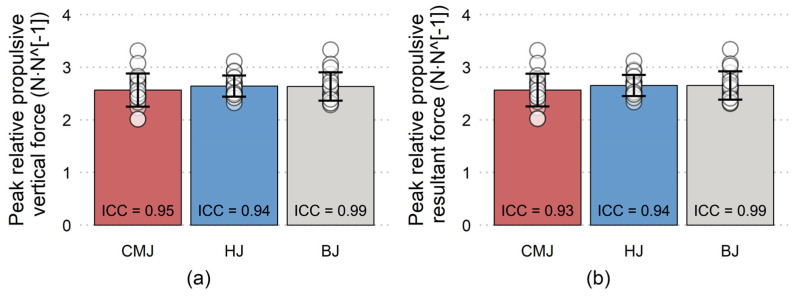
Mean (±SD) and inter-day interclass correlation (ICC) for (**a**) peak vertical propulsive force relative to bodyweight and (**b**) peak resultant propulsive force relative to bodyweight during countermovement jump (CMJ), hurdle jump (HJ), and box jump (BJ).

**Figure 4 jfmk-08-00061-f004:**
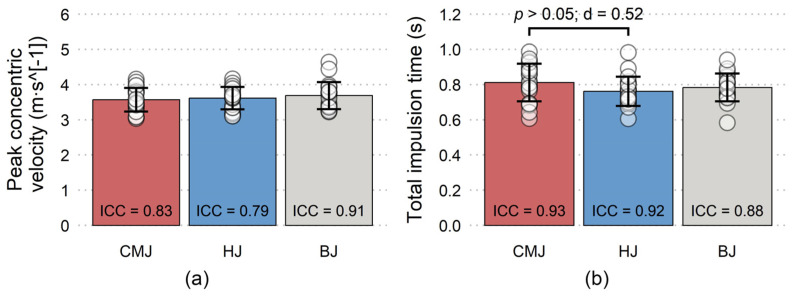
Mean (±SD) and inter-day interclass correlation (ICC) for (**a**) peak concentric velocity and (**b**) total impulsion time during countermovement jump (CMJ), hurdle jump (HJ), and box jump (BJ).

**Figure 5 jfmk-08-00061-f005:**
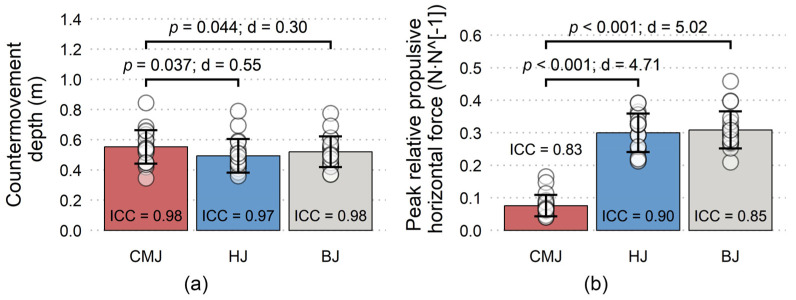
Mean (±SD) and inter-day interclass correlation (ICC) for (**a**) countermovement depth and (**b**) peak horizontal propulsive force relative to bodyweight during countermovement jump (CMJ), hurdle jump (HJ), and box jump (BJ).

**Figure 6 jfmk-08-00061-f006:**
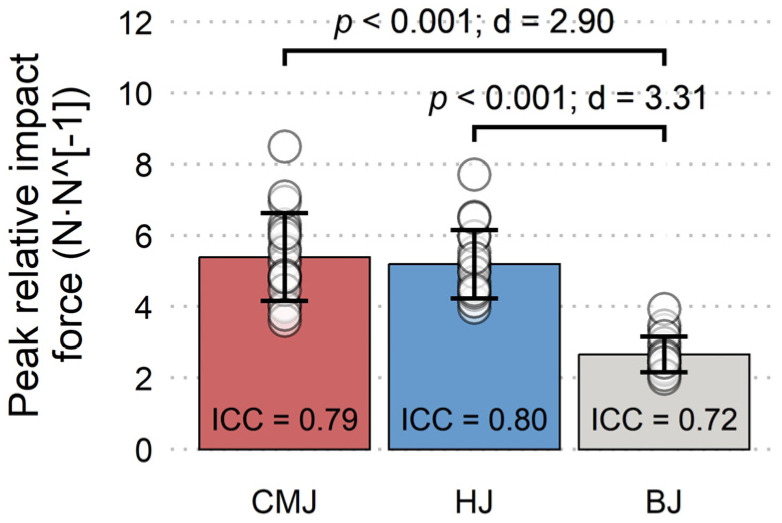
Mean (±SD) and inter-day interclass correlation (ICC) for peak resultant impact force relative to bodyweight during countermovement jump (CMJ), hurdle jump (HJ), and box jump (BJ).

## Data Availability

Data will be available upon a request sent to the corresponding author of this study.

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
