# Peer review of "Countermovement, Hurdle, and Box Jumps: Data-Driven Exercise Selection"

_jfmk, 2023, doi:10.3390/jfmk8020061_

Round 1

Reviewer 1 Report

Countermovement, Hurdle, and Box Jumps: Data-Driven Exercise Selection

In their study, the authors raised a very interesting issue of using the appropriate method of conducting three different plyometric exercises in training practice.

Therefore, from the point of view of coaching practice, the conducted research may be a very important indication for the appropriate selection of plyometric exercises depending on the sport discipline or training period.

The research procedure and the selection of the research group were presented correctly and quite extensively.

However, it is puzzling to present the results of the research, which include figures 1-3, while the rest of the results, Figure 4-8, have been presented in the "discussion" section.

This section requires efforts to systematize the "results" and "discussion" sections.

Reviewer 2 Report

The purpose of this study was to evaluate and quantify the differences of CMJ, HJ and BJ among the same subjects, in order to help coaches in the selection of different training regimen. The manuscription is clearly written, the experimental apparatus is advanced, and the experimental design is relatively completed, but the following problems still exist:

(1) While removing upper extremity effects from the study would provide more accurate parameters related to lower extremity jumping, does it make sense for real training practice since most jumps in daily training require upper extremity movements?

(2) In Line 104, it was mentioned that the subjects were required to place a light wooden dowel across their posterior shoulders behind the base of the neck for the experiment. Have you considered the effect of the subject's limited shoulder mobility for the subsequent experiment?

(3) 2.2 Data Collection component is excessive, to make the difficulty of reading. Suggest that this part could be subdivided into several parts such as test procedure and data collection, or create a simple experimental flow chart.

(4) The section of results is too short; it needs to expand some more specific research content.

(5) In Line 157, when testing a jump, will the jump be excluded if it is out of position, and will it be re-measured afterwards? In addition, are there other cases of failed jumps (e.g. failure to cross a hurdle or jump on a box) and what should be done in such cases?

Reviewer 3 Report

Introduction

Well written, with thorough critique of prior literature

lines 65-69: I recommend breaking these up into 2 or more sentences; in lines 68-69 you talk about both hurdle and box jumps, whereas previously you were only talking about box jumps, so it became a little confusing grammatically which statements applied to which jumps.

Methods

line 107: I'm confused about "time needed to perform the HJ and BJ"--so did they do a CMJ and then a HJ and BJ in those 8 seconds?

line 109: Revise for grammar and clarity: "a pilot testing showing no negative effects on following performance in studied population"

lines 120-126: So did they have the dowel racked on their posterior shoulders during the BJ and HJ?

data processing and stats methods appropriate to research design

RESULTS

line 183: PFr was never previously defined as an acronym; going back and forth I see that in line 141 you used PFh twice (which also helps explain my confusion in a comment I can now delete)

line 184: was the "moderate effect existed for longer TIT during CMJ..." also a statistically significant difference?

General comment: This suggestion you certainly don't have to follow if you want, I just want to throw it out there: I know we are all trained to maximize use of acronyms, but even in my own papers analyzing different force-time variables I cannot keep track of all of them. Since this is an open access journal with no word or page limit, consider significantly decreasing the number of acronyms you use so the reader doesn't have to try to remember each one. I would limit it to CMJ, HJ, BJ, and PF, with putting the full word for what is now the small letter (e.g. instead of PFv write PF-vertical).

Figures: instead of spreading all the figures out throughout the discussion, can they be moved to present sequentially before the discussion starts?

DISCUSSION

lines 223-225: This is one of the best practical explanations of kinematic performance variables I've seen in a research paper--well done

lines 283-285: In the methods, you described monitoring the dowel for problems, but in the results, can you please report how many trials you had to discount?

lines 289-290: Explain the technological constraints; it seems like for the box jump, if not the hurdle hop, attaching it to their waist would have still worked--there should have been more than enough height that the angle of the line would have cleared the top of the box

CONCLUSION

no comments

Reviewer 4 Report

Dear Authors

As one of the reviewers, I express my personal scientific opinion on your work. I would like to reassure you that I was trying to be positive and constructive but particularly as fair and honest as possible to your work. The clear explanation provided in Method’s section and the justification of the sample size using a-priori Power analysis are appreciated. I should also note that the originality of the study, the whole statistical approach used, the calculation of the Effect Size, the presentation of the ICC, the work done on figures are all positive points. I would expect however to see a clear section (i.e. Study Limitations) including the limitations of the study.

Please accept my judgment with a positive and constructive way.

Please see specific comments below:

Methods:

1.      Did you get any precaution concerning diet, caffeine ingestion and/or any other supplementation intake particularly prior to each official testing?

2.      Lines 158-159: You reported that “any trials with notable rotational movements or deviation from horizontal position were excluded”. In these cases, were the trials repeated by the participants?

Discussion/conclusion:

3.      I would expect to see a Study Limitations section.

Round 2

Reviewer 2 Report

It is appreciate that the authors submitted the revision according the comments.